# OpenReview forum: "Zephyr: Direct Distillation of LM Alignment"
_colmweb.org/COLM/2024/Conference — COLM_

### Official Review · Reviewer_T1tv · 2024-04-19

**Rating:** 7
**Confidence:** 4
**Ethics Flag:** 1

**Summary:**

This work produces a 7B large language model Zephyr, using distilled supervised fine-tuning and distilled direct preference optimization for alignment via AI feedback. Specifically, the authors first conduct dSFT on filtered UltraChat and dDPO on UltraFeedback (1-3 epochs) for LM alignment, based on the powerful Mistral 7B. Extensive experiments on different datasets and settings have been conducted, compared with various dPPO, RLHF, dSFT LLMs of different sizes. In conclusion, Zephyr 7B achieves impressive performance on most tasks.

**Questions To Authors:**

1)	More analyses on the dataset preprocessing of UltraChat/ UltraFeedback are encouraged to be released. For example, the authors randomly select one of the remaining three low-scoring responses of other models as the “rejected” response. Although the authors claimed that the randomness could improve the diversity and hardness of DPO. Is there any quantitative experimental analysis on this selection? Readers may want to know to which degree the diversity of low-scoring samples affects the performance of dDPO.
2)	This work only conducts dDPO based on Mistral 7B. It is interesting to verify the effectiveness of different LLMs.
3)	The authors have given some related works to preference alignment released after this work. The main differences between Zephyr and these works could be clarified.
4)	In Table 3, will adding more consideration on Safety in dDPO lower the performance of other aspects?

**Reasons To Accept:**

1)	This work explores the distilled direct preference optimization technique for LM alignment based on AI feedback, which could facilitate future research.
2)	The performance of the proposed 7B model is impressive.
3)	This idea is sound and easy to follow.
4)	Extensive experiments on different datasets and settings make the results more solid.

**Reasons To Reject:**

1)	The effectiveness of dDPO could be further verified on different LLM sizes and base LLM models.
2)	More details on the specific data preprocessing strategies on UltraChat and UltraFeedback should be discussed (better with more quantitative experimental analyses).
3)	More discussions on the side effects of considering safety in dDPO on other abilities (e.g., chat, reasoning in Table 3) should be given.

---

> ### Author Rebuttal · Authors · 2024-05-30
>
> Thanks for your thorough review and great questions. We were not able to run new experiments given the short time frame, but we can provide written answers to the main questions of interest.
>
> * Analysis of methods for converting responses to preferences.
>
> We agree that this would be a useful experiment to do fully quantitatively in practice. We do not have direct evidence that our sampling strategy is the most effective here, so we will temper this claim. Other works, such as NeuralChat, have used much simple strategy such as always assigning one model as strong and another as weak, and found this to be as effective as well.
>
> * Other LLMs
>
> While we present experiments in this work on Mistal-7B, other open source researchers have used the same recipe for Llama-70b (Tulu2), StableLM 3B (Stable Zephyr 3B), and Mistral 8x22B. Results seem to be relatively strong across different model classes. We will include a note about this in the paper and forward references to these works.
>
> * Preference Alignment
>
> This is a good question and we will survey this in the next draft. To summarize briefly: the main differences are: a) base model used, b) preference collection method, c) objective during training. (a) is covered above. (b) Refers to how prompts used to create the dataset are created. For example, different methods for using GPT-as-a-judge, different prompt filtering, methods that simply decide on a strong/weak model, and methods that use multi-round preference collection. (c) Refers to objective choices such as PPO style training, ranking style objective, and alternatives to DPO (KPO, IPO, SimPO) that use a similar setup but modify the loss.
>
> * Safety Training
>
> We hypothesize that safety training would likely lower some of the metrics. At the time of writing, it was not clear exactly how to do comparable safety training in a fair and measurable way for these models.

---

> > ### Comment · Reviewer_T1tv · 2024-06-03
> > **After rebuttal**
> >
> > Thanks for the authors' rebuttal, which has answered most of my questions. Considering the relatively high score I've given, I will maintain my voting of Accept.

---

### Official Review · Reviewer_4jHE · 2024-05-11

**Rating:** 4
**Confidence:** 4
**Ethics Flag:** 1

**Summary:**

This paper presents Zephyr which achieves state-of-the-art performance on chat benchmarks for 7B parameter models and outperforms the LLAMA2-CHAT-70B model. The model is built upon Mistral-7B with SFT on UltraChat and DPO on UltraFeedback.

**Reasons To Accept:**

- The Zephyr models are quite good. I believe this alignment recipe will benefit the open-source community.
- The experiments are solid. The authors provide sufficient experiment details and ablation studies, making the reproduction much easier.

**Reasons To Reject:**

- This paper is more like a technical report than a research paper. It does build a powerful open-source model, but there are limited research contributions. The core components, Mistral model, UltraChat, and UltraFeedback datasets, are proposed by others.
- The proposed terms dSFT and dDPO are unnecessary. They are still standard SFT and DPO, with merely different datasets.

Overall, I appreciate this paper for its empirical openness, including open-sourcing all code, data, model weights, and training recipes. However, the technical novelty is rather slim.

---

> ### Author Rebuttal · Authors · 2024-05-30
>
> Thanks for your comments. We are glad to hear that you found the approach useful.
>
> Let me attempt to persuade you of the contribution. We proudly acknowledge that work builds directly on Mistral, UltraChat, UltraFeedback, and DPO. However despite the ubiquity of Instruct-GPT, when this model was released the open-source community did not have any manner of inducing intent alignment behavior into models in a replicable and consistent manner. While the DPO algorithm was known, it had not been proven on anything but very small datasets. The main technical contribution of this paper is to bring these pieces together to make the first open-model with intent aligned behavior, as well as to show that DPO was sufficient for achieving that outcome. We show this by comparing it across many benchmarks, and demonstrate that the DPO stage is necessary to achieve that result.
>
> In retrospect the method is very simple. We're not trying to hide this. You do not need fancy RL or expensive annotations to produce useful models.

---

> > ### Comment · Reviewer_4jHE · 2024-06-03
> > **Re**
> >
> > Thanks for the reply. I agree that bringing pieces together is important, and the Zephyr recipe has been a standard approach for open-source alignment. However, the criteria in the open-source community and academia are different. Similar popular open-source models like LLaMA, Mistral, and Phi have never been published in conferences or journals. Therefore, since other reviewers already gave high scores, I insist my score to provide a different view.

---

### Official Review · Reviewer_Kgen · 2024-05-11

**Rating:** 7
**Confidence:** 4
**Ethics Flag:** 1

**Summary:**

This paper technically introduces how to produce a series of large language models within 7b parameters, based on open-source models, open-source distilled SFT and preference datasets. Comprehensive evaluations on different popular benchmarks are available.

**Questions To Authors:**

Do you have plans to try larger models?

**Reasons To Accept:**

1. The presentation is clear and easy to follow.
2. Detailed experiments and analysis. I believe the models and the paper are technically good contributions to the community.

**Reasons To Reject:**

It’s fine that Zephyr uses DPO for alignment and doesn’t have a PPO version. However, the authors denote that they compare “PPO approaches” in the experiments. I feel it’s a bit overclaimed since the dPPO model Xwin-LMv0.1 is based on a different model llama2.

---

> ### Author Rebuttal · Authors · 2024-05-30
>
> Thanks for your review.
>
> * Claimed comparison with PPO
>
> We agree that this claim should be modified and clarified. This should not be considered apples-to-apples
>
> We made a serious attempt to have a compatible PPO baseline, and in fact this was our original intention for the work. However in all experiments the PPO baseline performed worse than our DPO results and was significantly more costly. Given the complexity of PPO and challenges of replicating, we were unsure whether this was sound or due to a misconfiguration. Since submission, there do seem to be promising PPO results from other researchers.
>
> * Do you have plans to try larger models?
>
> Yes! In fact there are larger ones currently available (some from other labs).

---

> > ### Comment · Reviewer_Kgen · 2024-06-01
> > **Thanks for the reply**
> >
> > thanks for the rebuttal. Given the already high score, I’d like to keep it.

---

### Official Review · Reviewer_S1WR · 2024-05-18

**Rating:** 7
**Confidence:** 4
**Ethics Flag:** 1

**Summary:**

This paper proposes a training pipeline to achieve the performance of 40B+ scale large language models (LLMs) using a relatively small 7B model. Through three stages—dSFT, AIF, and dDPO—the authors transform Mistral-7B into the powerful Zephyr model, which outperforms various other 7B models. The effectiveness of the proposed training path is demonstrated through extensive performance comparisons with various LLMs.

**Questions To Authors:**

The paper can be seen as a form of imitation learning. While the significant improvement in chat performance is understandable, the substantial increase in academic benchmark performance shown in Table 2 is intriguing. What insights can the authors provide to explain this phenomenon?

**Reasons To Accept:**

- Proposes effective training pipeline to develop a small yet powerful model.
- Provides detailed explanations of the experimental setup.
- Conducts thorough performance comparisons with various models and meticulous ablation studies.

**Reasons To Reject:**

The paper lacks information on the cost involved in creating Zephyr. Details on the costs of using LLMs to create datasets in dSFT, generating answers with multiple models in AIF, and ranking responses using GPT-4 would help in understanding the practical feasibility of the proposed method.

---

> ### Author Rebuttal · Authors · 2024-05-30
>
> Thanks for the thorough review of the work. These are helpful comments.
>
> * Cost involved in creating Zephyr
>
> This is a good point, and we will include these results in an appendix.
>
> - Estimated cost of dataset construction: \$1.5k to judge with GPT-4
> - Cost of constructing roughly 250k alternative completions - 7 Hours on 8xH100 ~ $300
>
> While not cheap per se, these aspects are much less expensive than collecting high quality annotations at scale.
>
> * Imitation learning / Academic benchmarks
>
> This was also a bit surprising to us. The answer seems to be that the distinction between chat / academic is less clear cut than this terminology suggests. For example the UltraFeedback dataset includes completions from many prompts that are more academic than chat. By weighting the preference of GPT-4 preferred answer over a most likely uncorrect response, the model is getting signal for these tasks as well.

---

> > ### Comment · Reviewer_S1WR · 2024-06-05
> >
> > Thank you for your answer. I will maintain my score.

---

### Decision · Program_Chairs · 2024-07-10

**Decision:**

Accept

**Comment:**

Reasons To Accept:
1. The paper introduces Zephyr, a 7B large language model achieved through novel training methodologies dSFT, AIF and dDPO, demonstrating state-of-the-art performance on various benchmarks.
2. The experiments are extensive and well-documented, providing clear insights into the effectiveness of the proposed methods.
3. The approach addresses significant technical challenges in model alignment and performance improvement, contributing valuable insights to the community.
Reasons To Reject:
1. Some reviewers questioned the novelty of the contributions, noting that the core components (Mistral model, UltraChat, UltraFeedback datasets) were previously proposed by others.
2. There are concerns about the necessity and distinctiveness of terms like dSFT and dDPO, suggesting they are variations rather than fundamentally new methodologies.
3. Additional clarification is needed on the cost involved in creating and maintaining the Zephyr model, which was noted by reviewers as crucial for practical feasibility.

Overall, while the paper presents solid experimental results and technical advancements, however, the technical novelty is rather slim. I think that a paper with clear academic novelty can better attract readers and provide new insights.

[comments from the PCs] There's a lot to learn from the experiments in this paper. It would be very useful to get maximal transparency into the costs, especially as industry is becoming increasingly opaque in this regard.